# *ReSim*: Reliable World Simulation
# for Autonomous Driving

**Jiazhi Yang**[1,3]    **Kashyap Chitta**[4,7]    **Shenyuan Gao**[8]    **Long Chen**[5]    **Yuqian Shao**[6]
**Xiaosong Jia**[6]    **Hongyang Li**[2]    **Andreas Geiger**[7]    **Xiangyu Yue**[1†]    **Li Chen**[2,3†]

[1]The Chinese University of Hong Kong    [2]The University of Hong Kong
[3]OpenDriveLab at Shanghai AI Lab    [4]NVIDIA Research    [5]Xiaomi EV
[6]Shanghai Jiao Tong University    [7]University of Tübingen, Tübingen AI Center    [8]HKUST

https://opendrivelab.com/ReSim

## Abstract

How can we reliably simulate future driving scenarios under a wide range of ego driving behaviors? Recent driving world models, developed exclusively on real-world driving data with expert trajectories, struggle to represent hazardous or non-expert behaviors that are rare in training corpus. This limitation restricts their applicability to tasks such as policy evaluation. In this work, we address this challenge by enriching real-world human demonstrations with diverse non-expert data collected from a driving simulator (e.g., CARLA), and building a controllable world model trained on this heterogeneous corpus. Starting with a video generator featuring a diffusion transformer architecture, we devise several strategies to effectively integrate conditioning signals and improve prediction controllability and fidelity. The resulting model, **ReSim**, enables **Re**liable **Sim**ulation of diverse open-world driving scenarios under various actions, including hazardous non-expert ones. To close the gap between high-fidelity simulation and applications that require reward signals to judge different actions, we introduce a Video2Reward module that estimates a reward from ReSim's simulated future. Our ReSim paradigm achieves up to 44% higher visual fidelity, improves controllability for both expert and non-expert actions by over 50%, and boosts planning and policy selection performance on NAVSIM by 2% and 25%, respectively.

## 1 Introduction

Learning a world model capable of predicting plausible future outcomes is now envisioned as a key milestone in achieving autonomy [1, 2, 3]. Over the past decade, researchers have leveraged visual world models to learn compact representations [4, 5], guide test-time planning [6, 7, 8], and develop reinforcement learning agents [9, 10] across various domains [11, 12, 13]. Unlike general-purpose video generators, which prioritize visual fidelity and generalization, world models simulate futures with precise control over ego actions.

In the autonomous driving domain, recent driving world models have also made rapid improvements in visual fidelity and generalization by scaling to massive driving datasets [14, 15, 16] and integrating frontier video generation techniques [17, 18]. However, the ability to accurately follow actions, which is an essential requirement for precise reward estimation and effective planning [19, 20], remains challenging [21]. As real-world data continues to grow, a critical question emerges: *Is real-world human data alone sufficient to guarantee simulation reliability?* A notable limitation of real-world data is that it predominantly consists of safe *expert* demonstrations, where the state-action space is inherently restricted by safety and regulation concerns [22, 23]. Consequently, safety-critical or

---

†Corresponding authors. Primary contact to Jiazhi Yang at: jzyang@link.cuhk.edu.hk

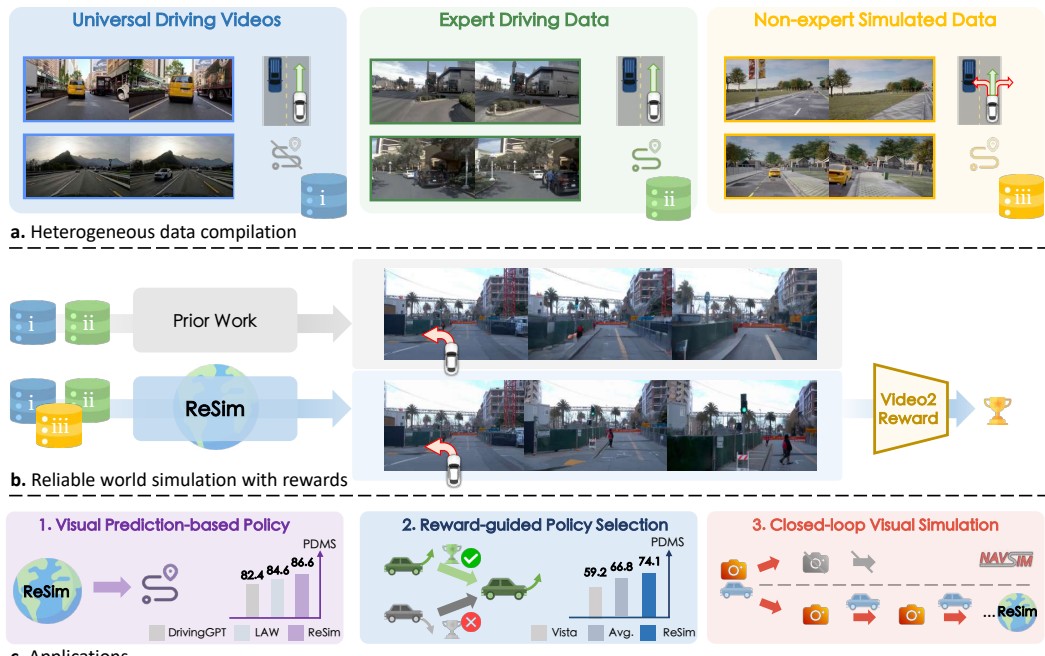

Figure 1: **Overview of ReSim.** **(a)** Heterogeneous driving data includes (i,ii) experts' safe driving logs, and (iii) potentially dangerous (non-expert) driving behaviors from simulations. **(b)** Prior driving world models are trained on expert data solely, leading to consistently safe yet inaccurate imaginations; in ReSim, we leverage all sources of data to simulate reliable and realistic futures, and build a robust reward model that generalizes to open-world scenarios within the simulator. **(c)** The high-fidelity prediction, accurate action-following, and reward estimation abilities of ReSim facilitate driving applications related to both policy deployment and simulation.

hazardous events (*e.g.*, collisions, off-road deviations) are significantly underrepresented [24, 25, 26]. This imbalance leads to severe hallucinations when the world model is exposed to unseen *non-expert* actions in certain states, undermining its robustness and reliability [27, 28].

To address the problem, we present **ReSim**, a reliable driving world model that can be steered by various actions, including out-of-distribution ones, while achieving high-fidelity simulation results. Our approach first enriches real-world human driving logs with non-expert data gathered from a driving simulator [29], where agents can execute a broader spectrum of actions without safety concerns. The resulting training corpus illustrated in Fig. 1(a) covers a wide spectrum of scenarios and actions (including non-expert ones), and further supports the simulation reliability for our world model. Built upon a scalable text-to-video generator [30], ReSim applies a multi-stage training pipeline to integrate visual and action conditions. We devise an unbalanced noise sampling strategy along with a dynamics consistency loss to emphasize the learning of motion coherence, especially when being applied to non-expert actions with significant visual changes. As showcased in Fig. 1(b), prior works like Vista [16] fail to follow the specified steering action, while ReSim accurately simulates off-road behaviors. Moreover, making the world model beneficial for real-world driving often requires reward estimation [1, 10, 31, 8], which judges the quality of different actions to guide decisions. Therefore, we develop a Video2Reward model to convert simulated video outputs from ReSim into scalar rewards in real-world scenarios.

Based on the above explorations, we further demonstrate applications for supporting real-world autonomous driving in various aspects, as depicted in Fig. 1(c). In scenarios where action conditioning is absent, the simulated future of ReSim can serve as a visual plan from which an executable ego trajectory can be derived. On the NAVSIM planning benchmark [23], with front-view sensory videos only, our video prediction-based policy achieves an improvement of +2.0 compared to state-of-the-art world model-based planners [32] and +2.6 compared to an end-to-end baseline [33] with supervised learning. Additionally, the integration of ReSim and the Video2Reward model offers a solution for selecting the trajectory with the highest estimated reward among those generated by candidate policies, thereby justifying and guiding the final decision. In our experiments, this policy selection

process leads to a performance boost of 55.3% in comparison to the weak candidate policies. More intuitively, our system offers a synthetic environment where we can validate the behavior of a learned driving policy by running it within the imagination of ReSim in a closed-loop manner.

**Contributions.** **(1)** While prior works either use simulated or real-world data separately to develop driving world models, we demonstrate that integrating both sources can alleviate the shortage of unsafe driving behaviors in real-world data, and can improve the model's action controllability in real-world scenarios. **(2)** We present ReSim, a controllable world model that reliably simulates high-fidelity future outcomes by precisely executing diverse action inputs, together with a comprehensive training recipe including an improved loss formulation and noise sampling strategy for incorporating condition inputs and capturing scenario dynamics. Rewards can be derived from the simulated futures via a Video2Reward model. **(3)** We applying ReSim to facilitate driving in real-world scenarios, and validate its effectiveness via benchmarking on a wide array of datasets and tasks, where it exhibits evident improvements over previous counterparts.

## 2 Reliable Driving World Simulation

We outline the ReSim framework as follows. In Sec. 2.1, we introduce the heterogeneous training data with a wide range of scenarios and actions. We instantiate ReSim on a diffusion transformer architecture with careful modifications to capture dynamic driving scenarios and enable accurate action conditioning in Sec. 2.2. We propose to derive rewards from the simulated results of ReSim via a Video2Reward model in Sec. 2.3. Furthermore, we demonstrate the applicability of our method in real-world driving applications in Sec. 2.4. More implementation details are in Appendix Sec. B.

### 2.1 Heterogeneous Data Compilation

Existing driving world models are typically developed on public autonomous driving datasets with expert trajectories [34, 35] and web videos [14]. Similarly, in this work, we compile these two sources, specifically NAVSIM [23] and OpenDV [14], within our training data. NAVSIM contains rigorously labeled actions for action control learning, while the large-scale OpenDV dataset supports generalization of the world model. However, as shown in Fig. 1(a), both sources are dominated by human behaviors. The lack of non-expert actions limits prior world models' ability to emulate non-expert behaviors and their corresponding outcomes, such as collisions. This issue further hinders the world models from effectively identifying an inferior driving policy and providing reliable rewards.

To address this limitation, we leverage a driving simulator, *i.e.*, CARLA [29], to gather data in synthetic environments, enabling exploration without the costs and risks associated with the physical world. Notably, although simulated data has been adopted for world models in synthetic environments [2, 10, 36, 37], such a source is overlooked in driving world models that operate in real scenarios [15, 14, 16, 38]. World models trained in simulation alone struggle to generalize to real world due to the significant visual gap between two worlds. Our data collection is conducted in CARLA with randomly sampled routes from Bench2Drive settings [39]. Two types of agents are deployed within the environments. One uses a well-established driving policy, PDM-Lite [40, 41], to collect expert executions, while the other adopts an exploration strategy whereby both the steering angle and the speed are randomly sampled from predefined sets to generate non-expert behavior data, which is underrepresented in human data yet a crucial component for training reliable world models. As a result, the numbers of video samples for each dataset are 4M for OpenDV, 85K for NAVSIM and 88K for CARLA. Each video sample is 4.9s long with a frequency of 10Hz, where the first 9 frames are visual context and the last 40 frames are prediction targets during training. See more data collection details in appendix B.1.

### 2.2 Controllable World Model

**Basics.**   ReSim is built on CogVideoX [30], a high-capacity diffusion transformer originally conditioned only on text. To enable visual-context and trajectory conditioning, we replace the denoiser's historical latent inputs with their clean counterparts (following Vista [16]) and project future ego waypoints via a learnable encoder into the transformer's input space alongside video

latents. The model is supervised by the following video diffusion loss:

$$\mathcal{L}_{\text{diffusion}} = \mathbb{E}_{\boldsymbol{x},\epsilon,t}\Big[\|\boldsymbol{x}^{k:} - D_\theta(\boldsymbol{x_t}; t, \boldsymbol{h}, \boldsymbol{c}[, \boldsymbol{a}])^{k:}\|^2\Big], \tag{1}$$

where $\boldsymbol{x}$ is the clean video latent, and $\boldsymbol{x_t}$ is the noised video latent constructed by imposing a randomly sampled noise $\epsilon$ on $\boldsymbol{x}$ at a diffusion timestep $t$. The diffusion transformer $D_\theta$ is conditioned on latents of historical frames $\boldsymbol{h}$, a high-level text command $\boldsymbol{c}$ (*e.g.*, "Turn left"), and a fine-grained action $\boldsymbol{a}$ which is a sequence of future ego waypoints. To focus on forecasting the future, the diffusion loss is applied to the latent from the $k$-th frame onward only, excluding the observed history.

**Dynamics Consistency Loss.** So far, the standard video diffusion loss (Eq. (1)) supervises each video frame independently, which overlooks temporal correlations in videos, resulting in inferior spatiotemporal coherence and realism [42, 43, 16]. To address this, we introduce a dynamics consistency loss to additionally supervise the "latent motion", the discrepancy of video latent across different timestep ranges. This loss forces predicted motion to match the ground truth. We compute this loss over multiple intervals to capture both short-term and long-term dynamics. The intuition behind this is that some agent behaviors, *e.g.*, yielding, are hard to capture in the short term. To stabilize the magnitude of the loss value, we further normalize this loss by a factor of $s$, which is the average value of absolute motion disparity for each interval. This loss is formulated as:

$$\mathcal{L}_{\text{dynamics}} = \mathbb{E}_{\boldsymbol{x},\epsilon,t}\Big[\sum_{j=1}^{K}\sum_{i=1}^{N-j}\frac{1}{s}\|(\boldsymbol{d}^{i+j} - \boldsymbol{d}^i) - (\boldsymbol{x}^{i+j} - \boldsymbol{x}^i)\|^2\Big], \tag{2}$$

where $\boldsymbol{x}$ and $\boldsymbol{d}$ are the ground-truth and model-predicted video latent respectively, and $i$ indexes the frame of the video latent. $K$ is the maximum timestep intervals considered for latent motion, which is set to 4 in our experiments. $N$ is the number of frames of video latent. The total loss for training the world model is the combination of video diffusion loss and dynamics consistency loss: $\mathcal{L} = \mathcal{L}_{\text{diffusion}} + \lambda\mathcal{L}_{\text{dynamics}}$, where $\lambda$ is set to 0.1 empirically. Note that both $\mathcal{L}_{\text{diffusion}}$ and $\mathcal{L}_{\text{dynamics}}$ are applied to the video latent compressed by the video VAE [30]. Therefore, the indices and number of frames in Eq. (1) and Eq. (2) correspond to the video latent representation instead of the raw video.

**Unbalanced Noise Sampling.** The behavior of a diffusion model is largely influenced by how we sample noise during training [44, 45], which controls how much noise is injected into the input data for the denoiser to recover [46, 47]. When applying commonly-used uniform noise sampling as in [30], we empirically find that our world model underperforms on complex driving dynamics, especially when we consider rare and non-expert behaviors. The issue behind this is that uniform timestep sampling lets models take a "shortcut" on low-noise diffusion timesteps where the model can recover the injected video noise by simply averaging information in adjacent frames, instead of learning critical motion details, which degrades the dynamics fidelity in generated driving videos [48]. To force the model to capture complex agent–environment interactions, we bias sampling toward higher-noise steps. We increase the frequency of drawing timesteps in [500, 1000] from $1/2$ to $2/3$, thereby amplifying input corruption and compelling richer dynamics learning.

**Progressive Multi-stage Learning.** We adapt CogVideoX [30], originally pretrained with text-only conditioning, into an controllable driving world model via a three-stage curriculum. **1)** We first endow it with the ability to predict futures that follow historical visual context and text commands, by training on OpenDV [14]. **2)** Next, we incorporate NAVSIM [23] and CARLA [29] with annotated actions for joint training with OpenDV. Action conditions, i.e., future ego waypoints, are encoded through a learnable transformer. Notably, NAVSIM trajectories are randomly masked ($p = 0.5$) to support both action-conditioned and free prediction, while CARLA waypoints remain intact to guide hazardous maneuvers that cannot be directly inferred from visual context. To prioritize structural dynamics over high-frequency details while improving training efficiency, we downsample inputs to 256×448, freeze the diffusion backbone, and fine-tune only the trajectory encoder and a LoRA adapter [49]. **3)** After the effective adaptation of action conditions, we finally resume the model training on 512×896 resolution with full fine-tuning, producing a model that generates 4s of 10Hz video conditioned on nine frames at 10Hz, an optional command, and a 4s, 2Hz waypoint sequence.

## 2.3 Reward Estimation from Video

To accomplish a feedback loop, world models need to estimate a reward to assess the predicted futures [1, 10], which is largely overlooked in prior driving world models [38, 14, 15]. Among the

few attempts, the lack of explicit goal states [19] in open-world driving and the complexity of outdoor scenarios make manual reward crafting challenging [31]. To overcome this, our key insight is to use the widely adopted simulator CARLA [29] as a rich source to learn rewards from via the unified video interface, as depicted in Fig. 2. Such a formulation offers several notable advantages. First, the driving simulator allows flexible exploration and can produce extensive data with environmental feedback to learn from. This includes not only successful driving experiences, but also non-expert mistakes and edge cases, covering a wide distribution of reward ranges. Second, contrary to constructing rewards manually with 3D perception models [31], the video interface does not require highly crafted 3D priors such as camera poses, and thus can benefit from a broad range of frontier vision models with strong cross-domain generalization [50, 51].

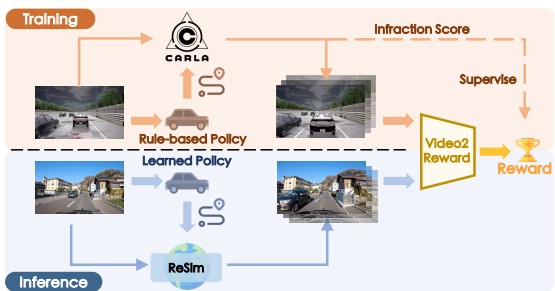

In detail, our Video2Reward model (V2R) is established on a frozen DINOv2 backbone [50] with an additional lightweight prediction head. Supervised by the CARLA infraction score [52, 29] that comprehensively penalizes multiple factors such as collisions and excessively low speeds, V2R learns to estimate the reward from video sequences. During inference, we send a planned trajectory produced by any policy to ReSim to simulate future video, which is then sent to V2R to estimate the reward of that trajectory. Due to the highly generalizable visual features of the DINOv2 backbone and the realistic prediction of ReSim, V2R is readily applicable to real-world driving scenarios, effectively assessing the quality of diverse behaviors.

Figure 2: **Video2Reward model (V2R). Top**: V2R is supervised by infraction score of both safe and hazardous data from simulation, deriving the reward from a driving video. **Bottom**: In real-world inference, the predicted video of ReSim in reaction to a proposed action is fed into V2R to estimate the action's reward.

## 2.4 Applications

**Video Prediction-based Policy.** From the future prediction capability learned from massive human driving videos at scale, ReSim implicitly learns how the ego vehicle should behave and can be converted into a video prediction-based policy, akin to recent approaches in robotics [53, 3, 54]. As opposed to solely imitating the ego trajectory, predicting future observations allows for utilizing a broader source of unlabeled video data while leveraging richer supervision, including the intention of surrounding agents that are not captured in sparse trajectory-based outputs. To serve as a policy for deployment, ReSim takes historical visual observations and a high-level command as input and imagines the unseen future images, without conditioning on actions (which should be the output of this task). After visual imagination, the predicted future frames of ReSim are fed into an inverse dynamics model (IDM) that converts it into a future trajectory of the ego vehicle. Illustrative samples are shown in Fig. 3, where critical events for ego planning are highlighted in dashed boxes.

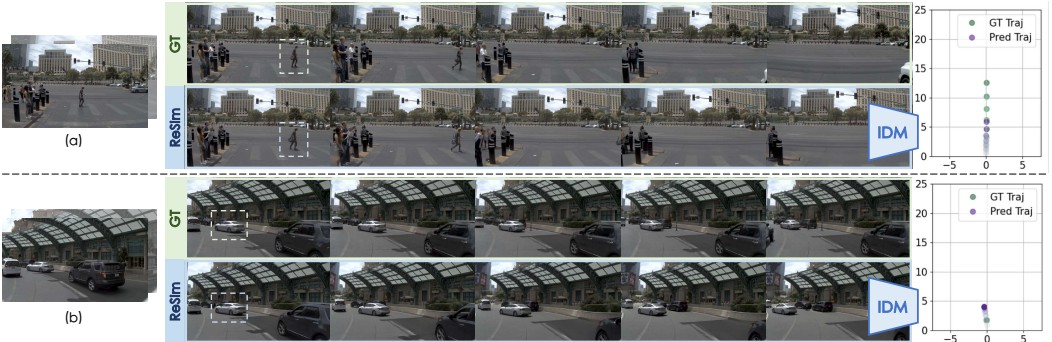

Figure 3: **Video prediction-based policy.** ReSim conditions on the history context (left) to synthesize a plausible visual plan (middle), which is then translated into an ego trajectory via an IDM (right).

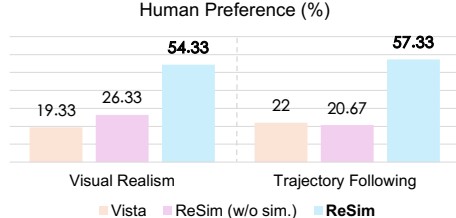

Human Preference (%)

Figure 4: **Human evaluation of non-expert action controllability.** ReSim gets the most votes in both realism and trajectory following.

Table 1: **Action-conditioned prediction accuracy on Waymo (zero-shot).** ReSim surpasses baselines by a large margin under both action conditions. w/o sim.: No simulation data for training.

| Method | Action-free ↓ | Expert Action ↓ |
|---|---|---|
| GT Future | 0.58 | 0.58 |
| Vista [16] | 5.68 | 1.89 |
| Ours w/o sim. | 1.47 | 1.18 |
| **Ours** | **1.13** | **0.86** |

**Reward-guided Policy Selection.** Complex driving environments often necessitate the maintenance of multiple candidate proposals to ensure planning robustness across various scenarios [55]. While it is straightforward to obtain multiple trajectory proposals from different policies for the same scenario, it raises the question of how to reconcile these diverse outputs. To address this, we propose to apply our method to score each trajectory with a reward and select the one with the highest reward for execution. Concretely, each candidate trajectory is rendered into a short predictive video using ReSim, and the resulting video is then passed through Video2Reward model to obtain its reward. Guided by the estimated reward, the trajectory selection process results in a steered policy with significant improvement over individual policy candidates, by leveraging their advantages in different situations.

**Closed-loop Visual Simulation.** Vision-based driving agents are primarily evaluated in an open-loop manner, either on static datasets against pre-recorded trajectories [34, 35] or simulation-based benchmarks that consider local interactions [23]. Both these evaluation types confine agents to safe and human-driven scenarios. More seriously, they overlook error accumulations over extended rollouts and fail to reflect the closed-loop performance as in real-world driving, where agents would be continuously exposed to new states after taking actions. Owing to its precise action controllability and high visual fidelity, we can leverage ReSim to simulate visual states in a closed-loop manner. In each iteration, ReSim executes the predicted action of the driving agent to generate the next visual state, which is then input to the agent to make decisions for the next iteration.

## 3 Experiments

In this section, we first evaluate ReSim's simulation reliability, specifically relating to its action controllability, video prediction fidelity, and reasonableness of the reward formulation (Sec. 3.1). Next, we validate ReSim's applicability to real-world driving tasks (Sec. 3.2). Finally, we present ablation studies on data and methodological designs to verify their effectiveness (Sec. 3.3).

### 3.1 Results of Simulation Reliability

**Results of Action Controllability.** We verify the action controllability of ReSim on the unseen Waymo Open dataset [35]. For action-free and expert action conditioning, we follow the protocol of Vista [16] and use the *Trajectory Difference* metric to assess how closely the world model's predicted future aligns with the input trajectory. As reported in Tab. 1, ReSim improves the results by 80% and 54% for both conditioning modes compared to Vista. Moreover, removing the simulated data from training (ReSim w/o sim.) results in a performance decrease for both conditioning modes. This evaluation is conducted on a random subset of the Waymo validation set with 540 samples.

For non-expert action conditioning, we conduct a human preference study among samples generated by different methods conditioned on non-expert actions. As reported in Fig. 4, ReSim outperforms baselines by a large margin for both visual realism and trajectory following. We also make qualitative comparisons between different methods in Fig. 5, where ReSim yields more reliable and realistic results that align with the non-expert trajectory input. Moreover, the learned action controllability can be transferred to unseen datasets in a zero-shot manner, as showcased in Fig. 6.

**Comparison of Video Prediction Fidelity.** The fidelity of video prediction is a key indicator of a driving world model's ability to simulate realistic scenarios. As presented in Tab. 2, we evaluate the

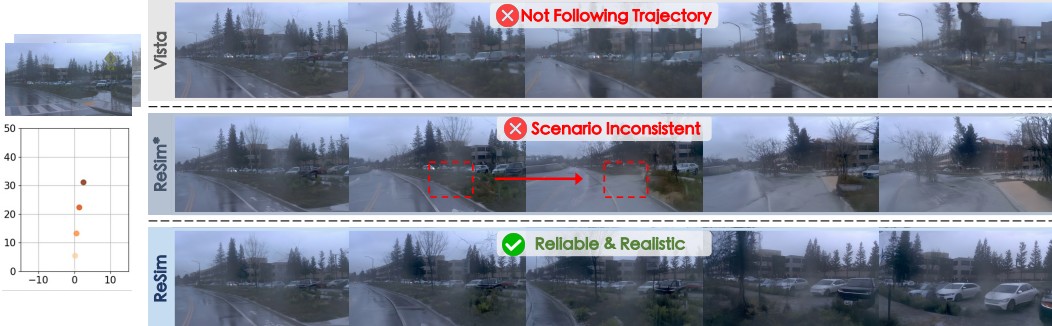

Figure 5: **Qualitative comparisons of non-expert action controllability.** ReSim reliably simulates hazardous outcomes from the non-expert action, while other methods either fail to follow the specified trajectory or compromise the scenario's consistency. ⋆: without simulated data in training.

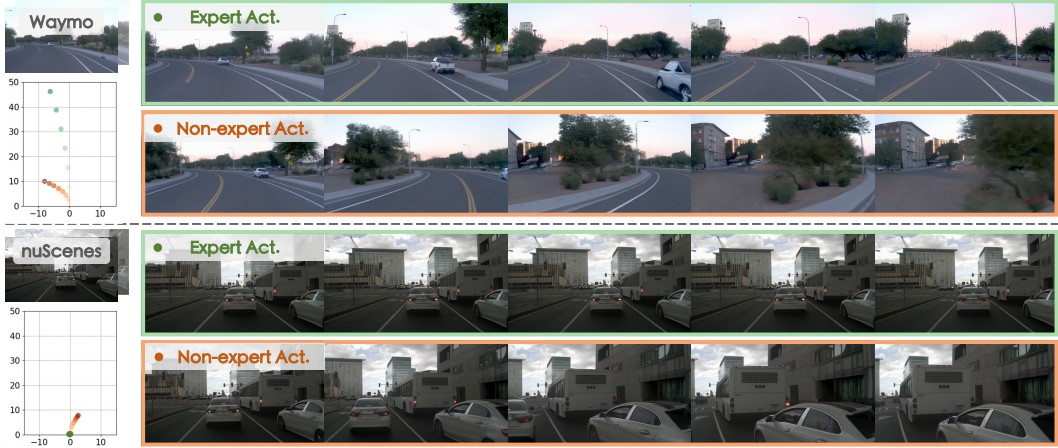

Figure 6: **Zero-shot action controllability.** ReSim can reliably follow both expert and non-expert actions in various scenarios from zero-shot datasets.

performance of various driving world models with FID [56] and FVD [57] metrics on nuScenes [34] validation set. The evaluation protocol follows Vista [16], using only context frames as conditions without imposing explicit action control. Notably, without training on any nuScenes data samples, ReSim yields significantly better results in a zero-shot manner compared to in-distribution models. We also provide qualitative comparisons for long-term future prediction in Appendix Sec. C, where

Vista's prediction becomes oversaturated and loses semantics of the scene, while ReSim remains predicting visually rich future states in 30s.

**Results of Reward Estimation.** To evaluate the effectiveness of our reward formulations, we measure the ability of each reward model to distinguish "expert" from "non-expert" trajectories via a reward correlation metric. Specifically, for both CARLA [29] and NAVSIM [23], we randomly sample successful episodes with expert trajectories and accompany each with a randomly drawn trajectory from other samples that is potentially unsafe and assumed as non-expert. Evaluation is conducted with 250 pairs of com-

Table 2: **Comparison of prediction fidelity on nuScenes validation set without action condition**. Without seeing any nuScenes samples during training, ReSim outperforms previous in-distribution driving world models.

| Method | Zero-shot | FID ↓ | FVD ↓ |
|---|---|---|---|
| DriveGAN [58] | ✗ | 27.8 | 390.8 |
| DriveDreamer [38] | ✗ | 14.9 | 340.8 |
| DriveDreamer-2 [17] | ✗ | 25.0 | 105.1 |
| WoVoGen [59] | ✗ | 27.6 | 417.7 |
| Drive-WM [31] | ✗ | 15.8 | 122.7 |
| GenAD [14] | ✗ | 15.4 | 184.0 |
| GEM [18] | ✗ | 10.5 | 158.5 |
| Vista [16] | ✗ | 6.9 | 89.4 |
| **Ours** | ✓ | **5.2** | **50.4** |

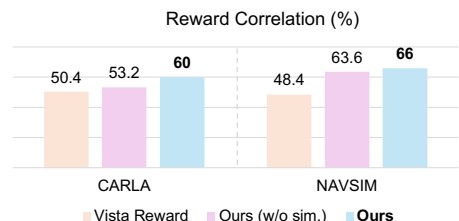

Figure 7: **Reward correlation.** Our method of composing ReSim and Video2Reward model yields more accurate rewards compared to baselines in both datasets.

Table 3: **Reward-guided policy selection.** Our reward formulation leads to a guided policy with +26% PDMS compared to candidate policies, outperforming other guidance and averaged ensemble.

| Method | Steered | PDMS ↑ |
|---|---|---|
| Transfuser [33] | ✗ | 47.7 |
| LTF [33] | ✗ | 47.2 |
| PDMS (Oracle) | ✓ | 94.2 |
| Average | ✓ | 66.8 |
| Vista [16] | ✓ | 59.2 |
| Ours w/o sim. | ✓ | 69.7 |
| **Ours** | ✓ | **74.1** |

Table 4: **Planning results on NAVSIM `navtest`.** ReSim outperforms both world-model (WM)-based planners and end-to-end (E2E) methods by a large margin, without accessing extra information.

| Method | Type | Multiple Sensors | Ego Status | Past Traj. | Extra Anno. | PDMS ↑ |
|---|---|---|---|---|---|---|
| VO planner [60] | E2E Plan | ✗ | ✗ | ✗ | ✗ | 78.4 |
| UniAD [61] | E2E Plan | ✓ | ✓ | ✗ | ✓ | 83.4 |
| Transfuser [33] | E2E Plan | ✓ | ✓ | ✗ | ✓ | 84.0 |
| DrivingGPT [62] | WM + E2E Plan | ✗ | ✗ | ✓ | ✗ | 82.4 |
| LAW [32] | WM + E2E Plan | ✓ | ✗ | ✗ | ✗ | 84.6 |
| GT Future (Oracle) | Ground-truth + IDM | ✗ | ✗ | ✗ | ✗ | 90.8 |
| **Ours** | WM + IDM | ✗ | ✗ | ✗ | ✗ | **86.6** |

parative samples for the reward model to judge. Reward models are expected to assign higher scores to expert trajectories compared to non-expert ones for the same scenario. Results in Fig. 7 validate the advantage of our method, which surpasses its counterparts in both simulated and real-world datasets.

## 3.2 Results of Applications

**Video Prediction-based Policy.** We evaluate the performance of our method on the `navtest` split of NAVSIM [23] benchmark. Specifically, we separately train an Inverse Dynamics Model (IDM) on the NAVSIM training set to convert the predicted video sequence of ReSim to an executable ego trajectory. As reported in Tab. 4, coupling ReSim and the lightweight IDM produces a video prediction-based policy that outperforms both end-to-end baselines (UniAD and Transfuser) and world model counterparts (DrivingGPT) by a non-trivial margin. Notably, our method only requires the history observations and a high-level command as input, without accessing multiple sensors, ego status, past trajectory, or extra annotations like other methods. The Visual Odometry (VO) planner shares the same architecture as our IDM yet performs poorly, underscoring ReSim's guidance.

**Reward-guided Policy Selection.** We compare different strategies for selecting an action from two candidate policies, *i.e.*, Transfuser and LTF [33]. The evaluation is conducted on a subset of NAVSIM, by selecting 300 challenging scenarios where one of the candidate policies fails while the other succeeds according to PDMS metric. As shown in Tab. 3, when applied separately, Transfuser and LTF achieve PDMS of 47.7 and 47.2, respectively. A uniform average ensemble lifts performance to 66.8, while the Vista reward only reaches 59.2. Instead, applying our reward strategy by composing ReSim and Video2Reward achieves a PDMS of 74.1, which is the closest score compared to the oracle selection according to ground-truth PDMS, and is higher than all baselines including our alternative (ours w/o sim.) that removes simulated data from the training of ReSim.

**Closed-loop Visual Simulation.** As showcased in Fig. 8, we leverage ReSim to iteratively simulate visual feedback for a running policy starting from two NAVSIM [23] scenarios. At each iteration, ReSim simulates an entire 4s future simultaneously by executing the action (*i.e.*, future trajectory for 4s) output by the policy. The newly generated frames are then fed into the policy for the subsequent

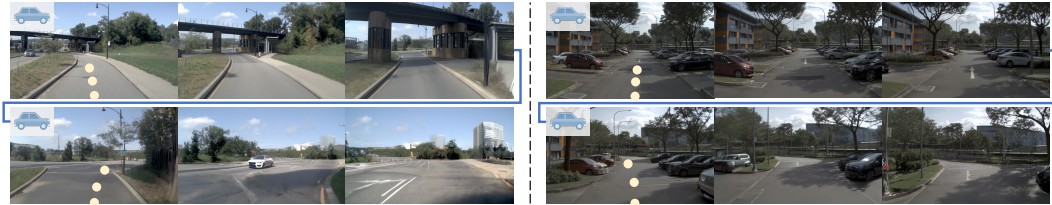

Figure 8: **Closed-loop visual simulation example.** A policy with front view only runs within the imaginary world generated by ReSim. The policy is adapted from XVO [60].

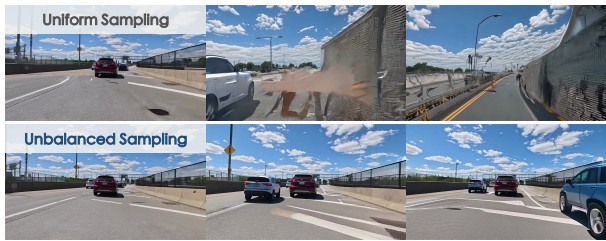

Figure 9: **Effect of unbalanced noise sampling.** Training with unbalanced noise sampling yields improved motion and scenario consistency.

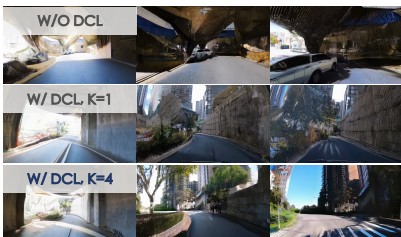

Figure 10: **Effect of dynamics consistency loss (DCL).** Applying DCL with $K = 4$ (in Eq. (2)) works best.

decision. We opt for a lightweight Visual Odometry-based planner adopted from XVO [60] as the policy, since it only takes front-view video as input. Attributed to the generative rollout, ReSim position the policy into states that are never encountered in a pre-recorded dataset.

### 3.3 Ablation Study

**Effect of Simulated Data.** Throughout our experiments, we demonstrate that training with simulated data improves results across multiple tasks. For action controllability, removing CARLA simulation data leads to inferior results for both expert (Tab. 1) and non-expert actions (Fig. 4). Without simulated data, the synthesized future may be inconsistent in the scenario's structure when conditioned on non-expert actions, as showcased in Fig. 5. Simulated data also contributes to more accurate reward estimates as shown in Fig. 7, which further benefits reward-guided policy selection (Tab. 3).

**Effect of Unbalanced Noise Sampling.** As shown in Fig. 9, applying unbalanced noise sampling during training makes the predicted future more consistent in terms of agents' motion and scenario layout, compared to the baseline with uniform noise sampling.

**Effect of Dynamics Consistency Loss.** We visualize the effect of applying our proposed dynamic consistency loss in Fig. 10. The qualitative results verify that incorporating the loss and extending the maximum interval $K$ for latent motion extraction (in Eq. (2)) yield more coherent future predictions.

## 4    Conclusion and Outlook

In this paper, we present ReSim, a reliable driving world model that excels in simulating a diverse range of actions in open-world scenarios. We incorporate non-expert data with hazardous actions from an established driving simulator to enrich real-world human driving data that primarily consists of safe behaviors. We also integrate several new training strategies, including a dynamics consistency loss, unbalanced noise sampling, and multi-stage learning. To facilitate driving applications beyond visual simulation, a Video2Reward model is devised to estimate the reward from the simulated future. Extensive experiments demonstrate the effectiveness and versatility of our ReSim system.

**Limitation and Future Works.** We envision our work as an early glimpse at open-world simulation with reward feedback, a cornerstone in establishing robust intelligence in the unstructured physical world. However, our system is still bottlenecked by inference efficiency due to iterative denoising, and

how to train agents within the synthesized world produced by ReSim is yet to be discovered. Future work focused on enhancing the efficiency, developing reinforced agents with the world model, and constructing fair closed-loop planning benchmarks would propel us closer to this goal. A discussion of limitations and broader impact of our work is included in Appendix Sec. D.

## Acknowledgments

This study is supported by National Natural Science Foundation of China (62206172) and Shanghai Committee of Science and Technology (23YF1462000). It is also supported in part by Centre for Perceptual and Interactive Intelligence (CPII) Ltd. (a CUHK-led InnoCentre under the InnoHK initiative of the Innovation and Technology Commission of the Hong Kong Special Administrative Region Government.), National Natural Science Foundation of China (Grant No. 62306261), the Shun Hing Institute of Advanced Engineering (SHIAE, Grant No. 8115074), and the EXC (number 2064/1 – project number: 390727645). This work is also partially supported by Hong Kong RGC Strategic Topics Grant STG1/E-403/24-N, and CUHK-CUHK(SZ)-GDST Joint Collaboration Fund YSP26-4760949.

We also thank the International Max Planck Research School for Intelligent Systems (IMPRS-IS) for supporting Kashyap Chitta. Our gratitude goes to Naiyan Wang, Shiyi Lan, Chonghao Sima, Haochen Tian, Yihang Qiu, Tianyu Li, Yunsong Zhou, and Qingwen Bu for valuable advice and discussions. We appreciate Huijie Wang's assistance in conducting the user study and constructing the project webpage.

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

# *Appendix*

In the Appendix, we first outline related works in Sec. A. We then demonstrate implementation details for data and models in Sec. B to supplement Sec. 2 in the main paper. Additional results are included in Sec. C to supplement Sec. 3 in the main paper. We discuss the limitations and broader impact of our work in Sec. D, and list the license of all assets in Sec. E.

## A    Related Work

### A.1   World Model

World models are considered as the abstraction of the open world, and having this kind of common sense greatly helps to learn new skills effectively, thus leading to high-level intelligence [1]. Under the definition of world models following the Dreamer series [10, 36, 37], they represent the transition of environmental dynamics, taking the past states or observations and policy's actions as input, and generating the next (latent) state together with an estimation of the reward. They also feature long-term prediction with continuous rollouts [2].

Abundant literature has explored world models in traditional policy learning tasks, especially utilizing the look-ahead property to learn efficient representations [63], conduct sampling-based planning [64, 65, 66], and enable model-based reinforcement learning [2, 10, 36, 37].

Taking a step further, researchers in applications have successfully employed world models in simulated games [67, 9, 11, 10, 36, 37], navigation [8, 12], and robotics [13, 68, 69, 3, 70, 71]. However, to learn and apply a world model requires extensive exploration and interaction with the environment, leading to the above advancements mostly being developed in simulation or constrained environments. It is infeasible to obtain diverse hazardous driving movements in the real world [72, 73]. In this work, for the first time, we address this challenge by leveraging heterogeneous data and transferring rewards learned from simulation to diverse real-world scenarios.

### A.2   Predictive Model for Driving Scenes

Driving Scenes are significantly unstructured, dynamic, and complex [74, 75, 76, 77], compared to standard policy learning environments such as Atari [78], DM Control [79], ViZDoom [80], *etc*. In order to effectively encode observations and facilitate restoring future environments, a wide span of representations have been explored to build the world state, including the vectorized representations [81, 82, 83, 84, 85], bird's-eye-view (BEV) representation [86, 87, 88, 89], point clouds [90, 91, 92], 3D occupancy [93, 94], images [58, 38, 17, 18, 95], and language [96, 97]. Meanwhile, these works mainly focus on public driving datasets, which are still limited in scales to achieve strong generalization ability. Inspired by the rapid growth of visual generative models [98, 44] and the increased data volume captured by cameras with low costs [15, 14, 99], recent world models that imagine future states in image sequence (*i.e.*, video) yield encouraging results in visual fidelity and generalization [15, 16, 18].

Unfortunately, prior methods still struggle to fulfill the mission of faithful simulation. Due to the insufficient learning of scenario dynamics, their imagination quality significantly degrades in challenging cases and long-horizon predictions [15, 16]. They also fall short in simulating negative consequences, such as car crashes, in response to bad ego actions, since they are mainly established on human driving logs, which are biased toward safe executions. Furthermore, the core problem for driving world models, how to deduce the reward for a given action and apply the world model for real-world driving problems, is largely understudied. In particular, with high-dimensional observations and complex relationships between agents and the environment, specifying rewards for open-world driving scenarios is challenging compared to goal-conditioned reward specifications [7, 65, 8]. Among the previous works, Wang *et al.* [31] propose to construct rule-based rewards with off-the-shelf 3D perception models [100, 101], yet these models are sensitive to sensor configurations like camera poses thus hard to generalize [102]. Uncertainty-based rewards in Vista [16] struggle to consider specific types of behaviors such as off-route actions. Our work meticulously investigates these challenges to facilitate planning and simulation.

### A.3 Video Generation

In recent years, deep generative models have made remarkable strides in both image generation [103, 98] and video generation [104, 44, 105, 106]. Recent studies [30, 107] introduce the diffusion transformer architecture [108] to video generation and achieve impressive spatiotemporal consistency. However, existing video generation models trained with large-scale web data are not directly applicable as driving world models due to their imperfect prediction of driving scenarios and lack of action controllability [14]. We bridge the gap with novelly designed model structures and training protocols.

## B Implementation Details

### B.1 Dataset

Our guiding observation is that each data corpus has distinct characteristics and limitations in terms of scenario diversity, planning labels feasibility, and the degree of danger, as depicted in Fig. 1(a). Based on that, we propose compiling our training data from diverse sources to integrate their complementary features to cover a wide scope of scenarios and ego actions. We specify each type of data source as follows.

**Universal Driving Videos.** Building a world model that generalizes to arbitrary scenarios requires learning from massive data with a wide coverage [13, 14, 99]. Therefore, we leverage the OpenDV dataset [14], which is the largest public driving video dataset, to pillar the scenario generalization of our world model. OpenDV dataset includes 1700 hours of uncalibrated front-view driving videos captured worldwide with a wide coverage of scenarios and camera configurations. The uncalibrated nature of this dataset allows the learned model to seamlessly adapt to new camera settings. We pseudo-labeled the dataset with high-level driving commands, including "`Turning left`", "`Moving forward`", and "`Turning right`", by estimating the flow via the OpenCV toolkit [109]. During training, we assign a high sampling rate ($5\times$) to video sequences with turning actions based on the driving command, as these cases are generally more challenging to learn than the forward movement. As a result, we collect 4M video clips from OpenDV datasets.

**Expert Driving Data.** Despite the large data volume and high diversity of online driving videos, these videos do not provide detailed annotations for ego actions, *e.g.*, ego trajectories, which are critical for learning world models with required action conditions [2]. The absence of such action annotations calls for the need to incorporate expert driving datasets that are rigorously curated and labeled. Therefore, we include a public driving dataset NAVSIM [23] into our compilation. We intentionally exclude commonly used nuScenes [34] and Waymo [35] datasets from training, and leverage them for held-out evaluation. Specifically, 85K data samples from `navtrain` split of NAVSIM [23] are included in training.

**Explorable simulated data.** Both online driving videos and expert driving datasets are produced by human drivers. The lack of suboptimal data would hinder the world model's ability to emulate non-expert behaviors and corresponding outcomes, *e.g.*, collisions. We randomly sample from 220 predefined routes in the Bench2Drive benchmark [39], varying the weather and time of day to enhance scenario diversity. We deploy two agents to explore the simulated environment while collecting data: One uses a well-established driving policy, PDM-Lite [41], to collect data from successful executions. Another agent for collecting non-expert data is implemented by rule-based explorations to cover a larger action space. This agent randomly samples a control configuration for steering angle and throttle and a behavior pattern from a predetermined set to execute. The total number of successful and hazardous execution cases is 88K, with each type accounting for roughly half the amount.

To be more specific about the 'non-expert' agent, it follows a structured "expert-takeover" process. First, the expert PDM-Lite policy drives for the initial period (1s). Then, control is switched to one of the following exploratory strategies to generate diverse, non-expert actions for 4s: 1) *Slight Turns*: The vehicle steers slightly left or right towards a randomly chosen angle between 10-20 degrees and then continues forward. 2) *Hard Turns*: The vehicle steers slightly left or right towards a randomly chosen angle between 10-20 degrees and then continues forward. 3) *Forced Lane Changes*: The vehicle executes a hard lane change to the left or right. 4) *Tailgating*: The vehicle disables its brakes

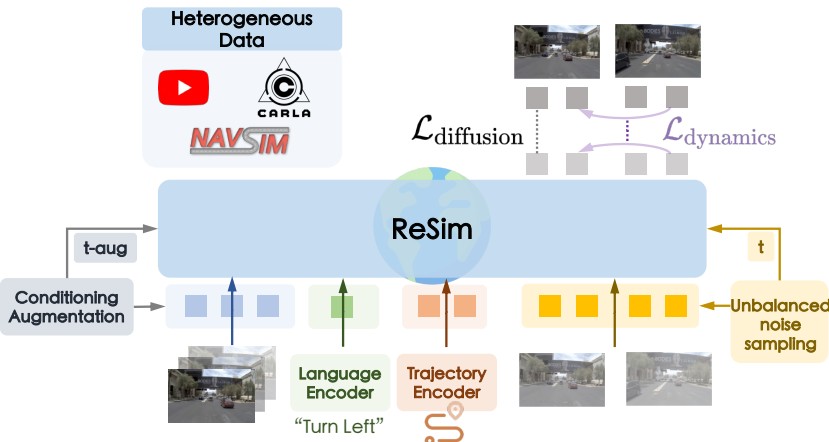

Figure S.11: **Overview of ReSim world model.** Learning from heterogeneous data compilation (Sec. 2.1), ReSim features designs specified in Main Sec. 2.2.

Table S.5: **Optimization configurations for different learning stages.** Traj. Enc.: Trajectory Encoder, Res.: Resolution, BS: Batch Size, LR: Learning Rate.

| Stages | DiT | LoRA | Traj. Enc. | Dataset | Res. | BS | LR | Steps |
|---|---|---|---|---|---|---|---|---|
| 1) | Trainable | - | - | OpenDV | 512×896 | 80 | $1e^{-5}$ | 20K |
| 2) | Frozen | Trainable | Trainable | OpenDV, NAVSIM, CARLA | 256×448 | 160 | $5e^{-5}$ | 80K |
| 3) | Trainable | Trainable | Trainable | OpenDV, NAVSIM, CARLA | 512×896 | 80 | $5e^{-5}$ | 50K |

and applies full throttle to induce a rear-end collision with a vehicle ahead. Each strategy also includes randomization of its internal parameters (e.g., turning angle, speed) to encourage action diversity.

## B.2 Model and Training

**ReSim World Model.** The architecture of ReSim is adapted from CogVideoX [30], consisting of a 2B diffusion transformer (DiT) as denoising backbone, a T5 encoder [110] for language encoding, a 3D Causal VAE that compresses raw videos into a compact latent space. Alongside language conditions for high-level driving command, we additionally devise a lightweight trajectory encoder, composed of two attention blocks and a linear head, to integrate the action condition into the DiT input. The overall architecture is depicted in Fig. S.11 with some of our designs highlighted. Besides our key innovations stated in Sec. 2, we also apply a conditioning augmentation strategy following [111, 11] to corrupt the video latent of historical observations to mitigate the error accumulation issue of long-term rollout. Similar to [11], the diffusion timesteps for historical context (t-aug) and future prediction (t) are separately sampled during training, while t-aug is always set to 0 during inference. This strategy improves the robustness of ReSim for multi-round prediction.

To enable classifier-free guidance for sampling [112], we randomly drop the textual command with a probability of $p = 0.5$. Similarly, we also drop the conditional ego trajectory at $p = 0.5$ for NAVSIM samples. However, we retain the ego trajectory for all CARLA samples without dropout, since their abnormal and hazardous behaviors cannot be accurately inferred from historical observations only and require explicit trajectory as guidance. Moreover, exposing the model to unconditioned hazardous behaviors could interfere with the learning of expert patterns from NAVSIM. Detailed learning configurations for different stages are included in Tab. S.5. All training stages are conducted on 40 A100 GPUs, and the total training duration is around 14 days.

**Video2Reward Model.** Video2Reward model consists of a pretrained DINOv2 [50] as backbone, and a prediction head that outputs a scalar reward. For each video sequence, all video frames are first processed separately via the image-based DINOv2 backbone. All image features are then passed to the prediction head, which aggregates all features via two consecutive spatial-temporal attention blocks and further predicts a scalar reward via an MLP.

Learning from our collected CARLA data only, Video2Reward model is supervised by the Infraction Score recorded from the CARLA simulator for each sample, which is a comprehensive evaluation of the ego driving performance [52] and penalizes behaviors such as collisions, traffic light violations, off-road deviations, and unreasonable low speed. It is trained for 20 epochs on a random subset of 35K samples from our CARLA data. We use the AdamW optimizer [113] with a learning rate of $1 \times 10^{-3}$. All video sequences are resized to 224×224 as input to this model.

**Inverse Dynamics Model.** Inverse dynamics model (IDM) estimates the ego trajectory from a video clip [114, 16]. Throughout our experiments, there are two parts that require the use of IDM, *i.e.*, the *Trajectory Difference* evaluation of expert action controllability (Sec. 3.1) and the application of video prediction-based policy (Sec. 3.2). These two IDMs are trained separately on different datasets, yet share the same architecture with a visual odometry backbone from XVO [60] and a lightweight attention head that outputs the ego trajectory with 8 waypoints in 2Hz.

For the *Trajectory Difference* of expert action controllability, the IDM transform model's action-control prediction into an estimated trajectory, and then we measure how closely the estimated trajectory matches the ground truth according to their L2 distance. A lower distance signifies a better action controllability of the driving world model. This IDM is trained on Waymo training set [35] for 40 epochs with a learning rate of $1 \times 10^{-4}$. For video prediction-based policy, the IDM transforms ReSim's action-free prediction (without command and ego future trajectory as condition) into an executable trajectory for planning. The IDM is trained on `navtrain` split of NAVSIM for 100 epochs. The learning rate for first 50 epochs is $1 \times 10^{-4}$ and decreases to $1 \times 10^{-5}$ for the last 50 epochs.

**Visual Odometry(VO)-based Planner.** The VO-based planner is utilized as a baseline for video prediction-based policy as in Tab. 4, and an agent that drives within the simulated world of ReSim for closed-loop visual simulation as in Fig. 8. It shares similar architecture and training to the aforementioned NAVSIM IDM. The only difference is that, instead of ingesting the whole video sequence containing both history and future frames as NAVSIM IDM, the VO-based planner takes historical frames as input only, without any explicit clue of the future observations.

## B.3 Sampling

With ReSim, each short-term future video is simulated by sampling with the DDIM sampler [115] for 50 steps. The simulated outcome is a 4s video sequence in 10Hz with a resolution of 512×896. The input conditions include 9 frames of historical observations in 10Hz, an optional high-level command, and an optional ego trajectory with 8 future waypoints in 2Hz. The high-level command is in one of "`Turning left`", "`Moving forward`", and "`Turning right`", and is classified either by estimated flow for OpenDV dataset [14] or ego trajectory for action-annotated datasets like NAVSIM [23] following common practice in [61, 116]. We always apply a prefix prompt, "`This video depicts a realistic view from the driver's perspective of a car driving on the road.`", concatenated with the textual command for both training and sampling. Empirically, this prefix helps guide the model to generate driving scenarios. Following CogVideoX [30], we apply a decreasing classifier-free guidance strategy with guidance scale starting from 7.5 and gradually decreasing to 1. To synthesize a longer future beyond the training horizon (4s), we can leverage the last 9 frames from the newly generated sequence as the context for next-round prediction iteratively. Simulating a 4-second video sequence takes two minutes on a single Nvidia A100 GPU.

## B.4 Human Evaluation

The human evaluation for non-expert action controllability (Sec. 3.1) is conducted with 15 participants and 40 questions for each participant, resulting in 600 answers in total. As showcased in Fig. S.12, each participant is requested to choose their preferred one among the synthesized video of three candidate models for each evaluation aspect. The candidate models are Vista [16], ReSim w/o simulated data, and ReSim (ours), and the evaluation aspects are Visual Realism and Trajectory Following. The association of different models and their generations is anonymous to participants.

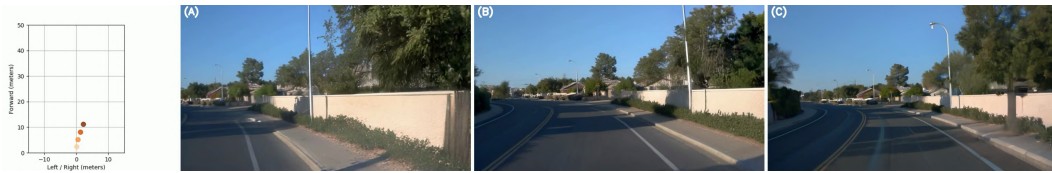

Figure S.12: **Example of human evaluation.** Participants are presented with synthesized videos of three anonymous candidate models. The order of different models' generations is shuffled for each testing scenario.

## C Additional Results

### C.1 Action Controllability

We provide additional visualizations for zero-shot action controllability in Fig. S.13 and Fig. S.14 for nuScenes and Waymo samples, respectively. Both datasets are unseen during training. Qualitative results demonstrate that ReSim can be flexibly controlled by both ground-truth trajectory (expert action) and randomly associated trajectory (non-expert actions).

### C.2 Ablation Study for Simulated Data

As shown in Fig. S.15, jointly training with simulated data improves the controllability of ReSim in open-world scenarios. Samples are from OpenDV validation set [14] with randomly associated trajectories from other labeled datasets.

### C.3 Action-free Prediction

We show the action-free prediction ability of ReSim in Fig. S.16. When conditioned on historical frames only without action inputs, ReSim synthesizes a possible future outcome, that might differ from the ground-truth due to the multi-modality of driving scenarios [117].

### C.4 Long-horizon Prediction

We compare ReSim with Vista [16] on long-horizon prediction in Fig. S.17. Starting from the same scenario, ReSim can emulate a more visually rich future in a longer horizon. This generation process does not use any action conditions, and both models perform multi-round rollouts that iteratively condition on the previously generated sequence to extend the prediction horizon.

### C.5 Failure Mode

Although ReSim exhibits improved fidelity and controllability over previous methods, it still faces challenges as in Fig. S.18. We discuss the limitation in Sec. D (Societal Impact).

## D Limitations and Broader Impact

**Inference Efficiency.** Despite the improved fidelity and controllability of our proposed ReSim, its real-world application is still potentially bottlenecked by the inference efficiency since diffusion models typically require multiple rounds of denoising process to ensure the generation quality [31, 44, 16]. To improve the inference latency, one potential solution is to reduce the number of denoising steps during the sampling phase. Recent advances in robotics [118] have proven that even with a single forward pass of the generative denoising network, the produced representation would greatly benefit downstream planning performance. Another approach is to distill a large yet slow diffusion model into a smaller one, which can be real-time deployed [119, 120].

**World Model for Policy Training.** Besides the onboard deployment of the heavy world model, another promising direction is to apply the world model as an dynamic environment to train policies [2, 10, 121]. This is beneficial as we can then deploy the policy to the autonomy directly, instead of

the world model, upon the training convergence of the policy model. Inspired by the tremendous success of large-scale policy learning within the abstract simulator without visual signals [122], the proposed ReSim offers a great opportunity to reproduce and go beyond the human-level robustness in the regime of vision-based driving [61, 123] by scaling up ReSim's visual simulation. We will follow this research direction in future work.

**Closed-loop Benchmark.** As illustrated in the results in Sec. 3.2, ReSim can reactively expose the policy to new states beyond the human driving logs when serving as a closed-loop visual simulator, in contrast to current predominant evaluation benchmarks for end-to-end autonomous driving [34, 35, 23]. However, since ReSim is trained on front-view observations only, common planning methods with multi-view camera inputs, such as UniAD [61] and VAD [123], cannot be readily applied in such simulation. Moreover, how to fairly benchmark different policies quantitatively using ReSim is still worth exploration.

**Societal Impact.** Though meticulously developed with state-of-the-art performance shown in the results, ReSim might still exhibit uncontrollable visual artifacts in generation due to the stochastic nature of the diffusion framework. It might also hallucinate in complex scenarios with multiple agents involved, and further pose risks for downstream applications. Despite the training on large-scale datasets, the uncurated data distribution, such as geographical regions, might lead to biased behavior of the learned model. We hope our work could shed light on the construction of open-world neural simulation for physical intelligence spanning both driving and robotics, by leveraging the visual richness of the physical world and the action flexibility of the simulated world collectively.

# E  License of Assets

Our training and evaluation are conducted on publicly licensed datasets and benchmarks [34, 124, 35, 23, 14]. To improve action diversity, we collected some data from the CARLA simulator [29] under the CC-BY License. The scenario configurations for the CARLA data follow Bench2Drive [39] under CC BY-NC-SA 4.0. ReSim is developed upon CogVideoX [30], with both code and model under the Apache License 2.0. We adopt public visual encoders, including DINOv2 [50] (under Apache License 2.0) and XVO [60] (under CC BY-NC-SA 4.0) for the construction of our Video2Reward and inverse dynamics model, respectively. Vista [16] is leveraged as a comparative baseline, which is under Apache License 2.0. We will release our code and models under the Apache License 2.0.

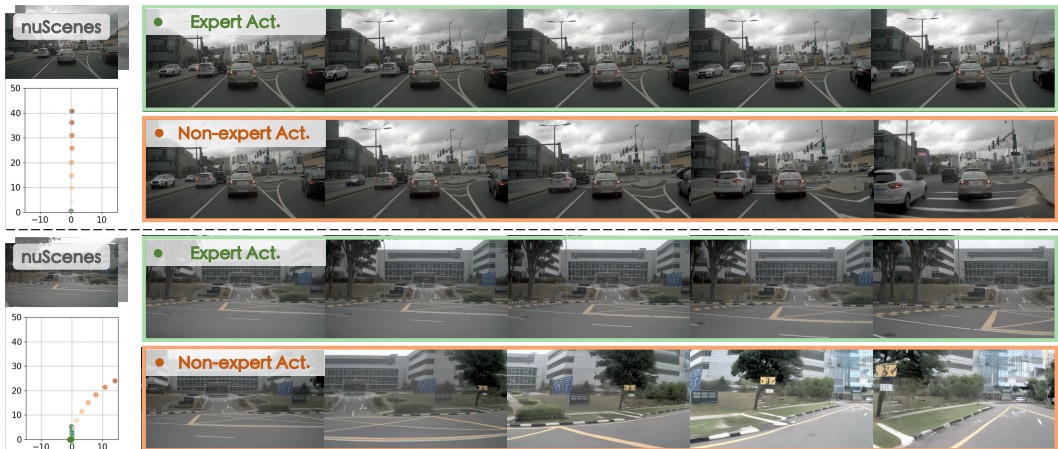

Figure S.13: **Visualizations for zero-shot action controllability on nuScenes.** The expert actions are recorded ground-truth from the driving log, while non-expert actions are randomly sampled from other scenarios. Best viewed zoomed in.

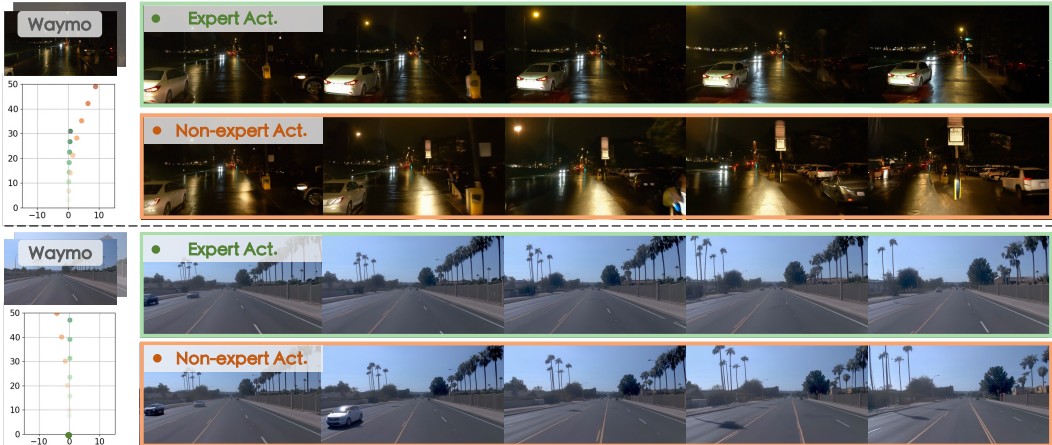

Figure S.14: **Visualizations for zero-shot action controllability on Waymo.** The expert actions are recorded ground-truth from the driving log, while non-expert actions are randomly sampled from other scenarios. Best viewed zoomed in.

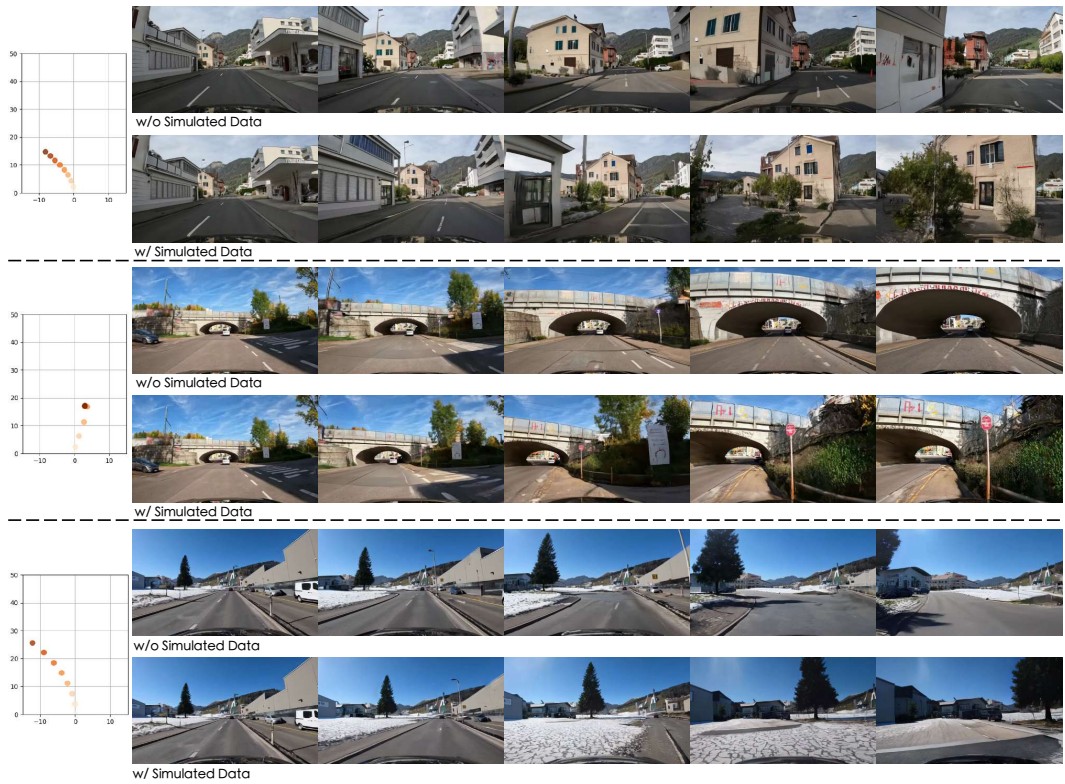

Figure S.15: **Additional ablations for incorporating simulated data in training.** Simulated data improves controllability of ReSim for non-expert actions. Historical frames are not shown for brevity.

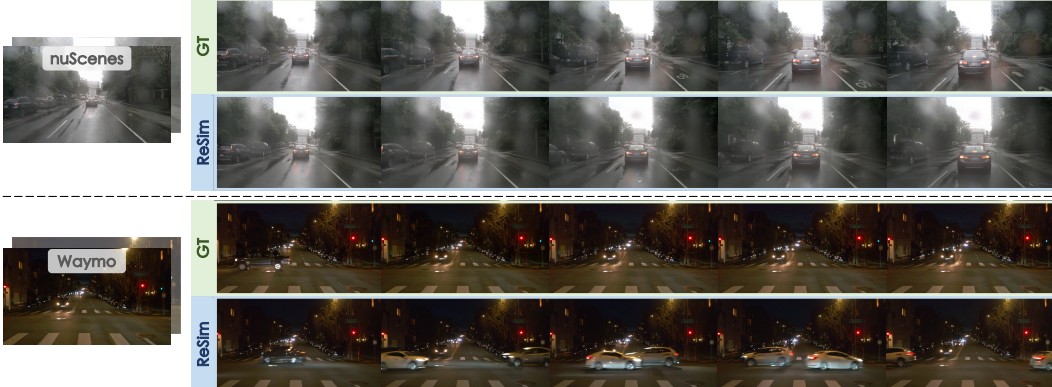

Figure S.16: **Visualizations for action-free future prediction.** ReSim can predict the future without action conditions by inferring from historical frames only.

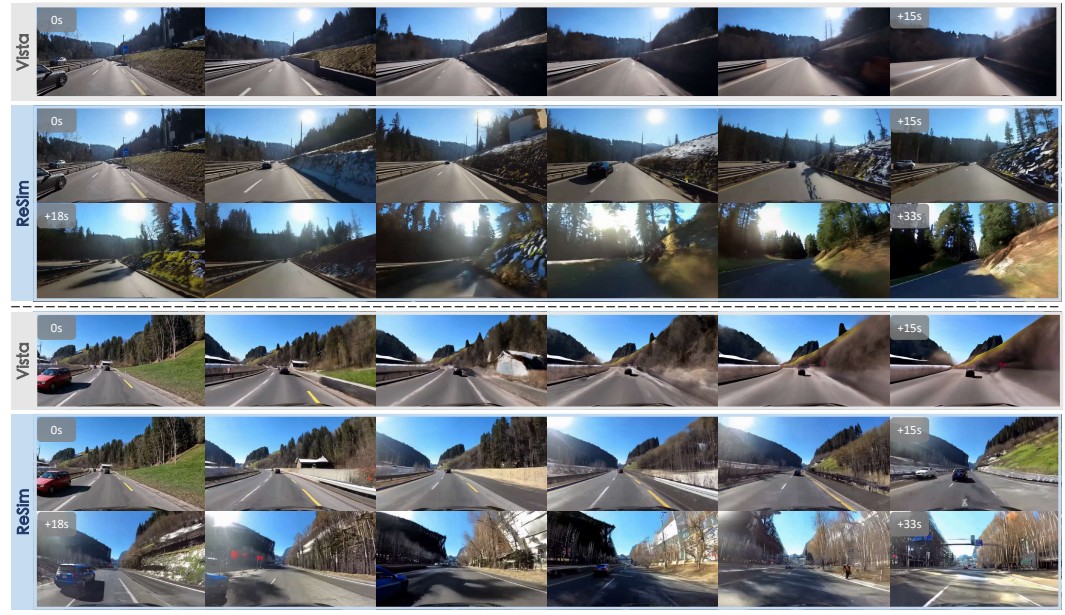

Figure S.17: **Long-term future prediction.** Compared to Vista whose prediction fidelity severely degrades in 15s, ReSim can predict consistent future states with rich details in more than 30s.

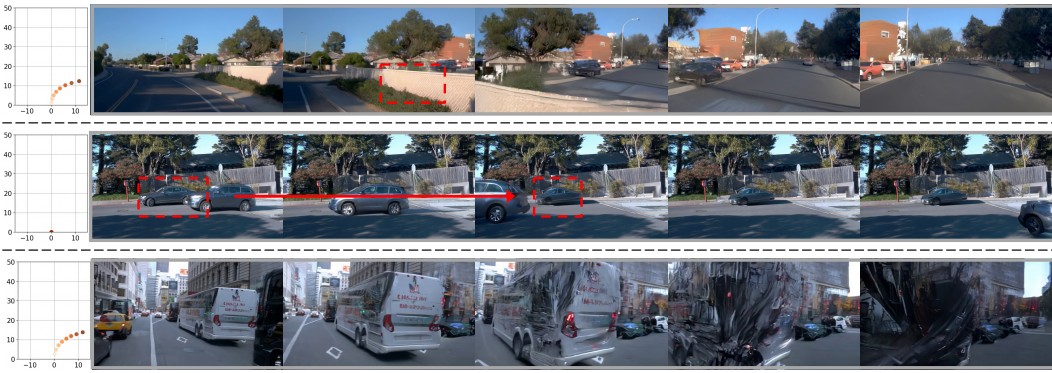

Figure S.18: **Failure modes.** ReSim still struggles in certain scenarios, such as falsely crossing the parapet, poor consistency for occluded objects, and producing visual artifacts for extreme cases. Best viewed zoomed in.

