# OpenReview forum: "ReSim: Reliable World Simulation for Autonomous Driving"
_NeurIPS.cc/2025/Conference — NeurIPS 2025 spotlight_

### Official Review · Reviewer_Wvvy · 2025-06-25

**Clarity:** 3
**Significance:** 3
**Originality:** 3
**Rating:** 5
**Confidence:** 2

**Summary:**

This work aims to address the lack of non-expert data in training the driving world model. It collects non-expert data from the driving simulator CARLA. Combining such data with real-world driving videos and simulated expert driving data, the authors trained a controllable world model, ReSim, which reliably simulates high-fidelity future outcomes. Experiments in real-world scenarios exhibit the advancements of the proposed method.

**Questions:**

1. As this paper adapts CogVideoX, do other baselines also use the same video model?
2. Following weakness 1, what are the contributions of expert and non-expert simulation data?

**Ethical Concerns:**

["NO or VERY MINOR ethics concerns only"]

**Final Justification:**

I think this paper is technically solid, and my minor concerns are addressed by the authors in the rebuttal. I will maintain my score as Accept.

**Limitations:**

See weakness and questions. No negative societal impact.

**Paper Formatting Concerns:**

No paper formatting concerns.

**Quality:**

3

**Strengths And Weaknesses:**

Strengths:
1. I think the lack of non-expert data in training driving world models is a real problem and collecting them through CARLA is also an efficient way. Meanwhile, the heterogeneous data compilation provides diverse and reliable data for training.
2. The proposed unbalance noise sampling and dynamic consistency loss are inspiring and validated through ablation studies.
3. The supplementary material includes multiple demo videos.

Weakness:
1. The ablation study includes the effect of simulated data. However, as one of the main contributions of this work is the introduction of non-expert data included in training, I wonder what this part alone contributes to the final performance?
2. I don't see an obvious novelty in the model design. However, I think the contribution to the non-expert driving data is good enough.

---

> ### Author Rebuttal · Authors · 2025-07-30
>
> Thank you for the careful and constructive review. We answer each of your questions as follows, and will update these discussions in the revision.
>
> > ### Effect of expert and non-expert simulation data alone.
>
> Our intuition of including both expert and non-expert simulation data is to cover the wide range of possible behaviors. Non-expert simulation data is generated by predetermined rules (such as slight turns and hard turns), without much consideration of its likeness to human behavior. Thereby, we also include expert simulation data to compensate for the loss of human-like maneuvers such as driving smoothness. We agree that explicitly ablating their effects would be insightful and will consider it in future experiments.
>
>
> > ### Inobvious novelty in model design, yet good data contribution.
>
> As you pointed out, our primary contribution is a data-centric methodology that incorporates heterogeneous data from multiple sources to train a reliable driving world model. The essence of the proposed method is our tailored training recipe (Sec. 2.2), which includes a new dynamics loss, improved noise sampling, and a multi-stage learning scheme for effective learning on heterogeneous data, which could be insightful for future work in this direction.
>
> > ### Do baselines use the same video model?
>
> We primarily use two models as our baselines, namely ReSim (w/o sim.) and Vista.
>
> * ReSim (w/o sim.) baseline uses the same base model (CogVideoX [1]), and training procedure as our full model; the only difference is the exclusion of the CARLA data. This provides a fair ablation of the contribution of the simulated data.
>
> * Vista is a widely used baseline, which is developed upon the Stable Video Diffusion [2] with different training data and procedures. The comparison against Vista demonstrates the overall improvement of our proposed system.
>
> We will clarify their architectural distinction in the revision.
>
> ---
>
> [1] CogVideoX: Text-to-Video Diffusion Models with An Expert Transformer
>
> [2] Stable Video Diffusion: Scaling Latent Video Diffusion Models to Large Datasets

---

> > ### Comment · Reviewer_Wvvy · 2025-08-03
> >
> > Thanks authors for the rebuttal. I don't have any additional questions. I will maintain my rating as Accept.

---

> > > ### Author Response · Authors · 2025-08-03
> > >
> > > Thanks for the reply. We appreciate your help in improving our work.

---

### Official Review · Reviewer_SA1m · 2025-07-02

**Clarity:** 4
**Significance:** 3
**Originality:** 3
**Rating:** 5
**Confidence:** 4

**Summary:**

The paper proposes an approach of learning driving behavior from synthesizing diverse world driving models combining real-world datasets and synthetic data. Their approach is built on a diffusion transformer model, by conditioning on frame latent features, high-level text command and action given by a set of future waypoints. They provide regularization based on dynamic consistency, bias sampling towards higher-noise steps and apply curriculur learning to train the model. For the reward to assess predicted futures, they turn towards CARLA using a infraction score to drive the model towards the best learned policy. They show that this paradigm achieves greater visual fidelity and controllability for expert and non-expert actions.

**Questions:**

Written in weakness section

**Ethical Concerns:**

["NO or VERY MINOR ethics concerns only"]

**Final Justification:**

The authors have satisfied with their reasoning and rebuttals.

**Limitations:**

Written in limitations section

**Quality:**

3

**Strengths And Weaknesses:**

Strengths -
- Combining real-world data and simulation data for diversity in driving behaviors
- Performing Video2Reward using Carla infraction score makes the evaluation criteria homogeneous between real and sim
- Showcases controllable simulator and provides prediction fidelity
- It performs planning without accessing extra annotation
Weakness -
- There is a large amount of domain shift whenever real and simulated data is mixed. However, the authors do not mention how they tackle the problem other than a single sentence mention (line 96).
- They provide the ratio of samples of OpenDV (4M), NAVSIM (85K) and CARLA (88K). Why are these ratios chosen - what happens when the ratios are switched?
- If the same scenario (single lane driving) has two different actions, will the algorithm be biased towards the one it has seen more of (weighted more)? Is there a way the authors use towards debiasing the inputs?
- How much noise can the algorithm tolerate during training and inference?

---

> ### Author Rebuttal · Authors · 2025-07-29
>
> Thanks for your thoughtful review. We address each of your concerns below and will include all discussions in the revision accordingly.
>
> > ### Details on how to handle the domain shift among different corpus.
>
> We mitigate the domain shift between real and simulated data through three key strategies:
>
> * Dominance of Real Data: Our training corpus is overwhelmingly real-world data (4M+85K real samples vs. 88K simulated). This grounds the model's visual understanding in reality, while the small CARLA dataset is primarily used to teach the dynamics of rare events.
>
> * Progressive Multi-stage Learning: We first train the model exclusively on real data. We then introduce mixed data while freezing the model's backbone and fine-tuning only a lightweight LoRA adapter, which prevents catastrophic forgetting of real-world visuals.
>
> * Improved Dynamics Modeling: Our Dynamics Consistency Loss (Sec. 2.2, lines 116-131)  encourages the model to learn abstract motion patterns instead of just visual details, making the dynamics learned from simulation more transferable to the real world.
>
>
> We will elaborate on this part in the revision.
>
> > ### Intuition behind different data scales.
>
> The intuition of different data scales is detailed in the following:
>
> * OpenDV (4M): We use all training samples from filtered OpenDV following Vista to maximize scenario generalization.
>
> * NAVSIM (85K): We included all samples from the NAVSIM training split to provide expert trajectories for learning controllability.
>
> * CARLA (88K): We collect this data of a similar size to NAVSIM to provide balanced action-controlled samples from the real-world and simulated domains. Despite its small scale, this data effectively improves controllability and reward estimation.
>
> We did not perform tuning their ratio due to computational constraints. We believe adding more labeled expert and non-expert action data would further improve controllability, which is left as future work.
>
> > ### Data debiasing to enable controlling diverse actions.
>
> According to Sec. 2.1 (in lines 99-101), we debias the data collection process for non-expert actions via explorative driving strategies. For each data sample, one of the following strategies is randomly chosen for the ego vehicle:
>
> * Slight Turns: The vehicle steers slightly left or right towards a randomly chosen angle between 10-20 degrees and then continues forward.
>
> * Hard Turns: The vehicle performs a sharp left or right turn towards a random angle between 40-60 degrees.
>
> * Forced Lane Changes: The vehicle executes a hard lane change to the left or right.
>
> * Tailgating: The vehicle disables its brakes and applies full throttle to induce a rear-end collision with a vehicle ahead.
>
> Each strategy also includes randomization of its internal parameters (e.g., turning angle, speed) to encourage action diversity. We will add these details to the revision.
>
> > ### How much noise can the algorithm tolerate during training and inference?
>
> We hypothesize your question is related to the noise in diffusion process. We employ an Unbalanced Noise Sampling strategy (Sec. 2.2, lines 132-142) that biases training towards higher-noise diffusion timesteps (timesteps 500-1000 out of 1000). This forces the model to learn the underlying data structure and dynamics from highly corrupted information, rather than taking "shortcuts" on low-noise examples that are slightly corrupted and can be easily denoised. This approach improves robustness and dynamics, as shown qualitatively in Figure 9.

---

> > ### Comment · Reviewer_SA1m · 2025-08-01
> > **Further discussion**
> >
> > **Domain shift**
> > Our experience in domain adaptation is that there is a directionality in domain adaptation - it moves towards the domain it is trained on later even if the data size is smaller. In your case, if you are fine-tuning on CARLA data later and freezing the diffusion backbone - how can you assure that it does not learn the non-expert behavior and forget the expert behavior?
> >
> > **Data debiasing**
> > It seems that you are using a very small and simplified class of non-expert actions. As such, the question becomes whether it spans the entire distribution of non-expert behaviors?
> >
> > **Noise**
> > I do not mean the sampled noise but the noise from the assimilated videos and expert data? Possibly the noise stemming from the simplistic dynamics of CARLA as well.

---

> > > ### Author Response · Authors · 2025-08-03
> > > **Reply to further discussion**
> > >
> > > Thank you for the detailed and thoughtful follow-up. We address each concern below.
> > >
> > > > ### Domain shift.
> > >
> > > To reduce the risk of catastrophic forgetting, the adaptation phase (Stage 2 & 3) does not use CARLA data only, but uses a mixture of all data including OpenDV, NAVSIM and CARLA, as shown in the table below (please refer to Table S.1 in Supplementary PDF for more details).
> > > Recall that the model’s task is conditional prediction: p(future_frames | past_frames, input action). The co-training strategy on all data corpus encourages the model to follow the input action reasonably for both expert and non-expert cases.
> > >
> > > | Stages | DiT  | Lora   | Dataset                |
> > > |:-------|:---------- |:---------- |:-----------------------|
> > > | 1)     | Trainable  | - | OpenDV                 |
> > > | 2)     | Frozen   | Yes | OpenDV, NAVSIM, CARLA  |
> > > | 3)     | Trainable  | Yes | OpenDV, NAVSIM, CARLA  |
> > >
> > > > ### Data debiasing.
> > >
> > > It is important to note that each of these patterns is a distribution of actions. The parameters within each primitive, such as the precise turning angle, speed, and duration, are randomized for every data sample. Therefore, the model is exposed to a much wider variety of trajectories than these sparse labels suggest.
> > >
> > > Furthermore, due to the diverse environmental layout and stochastic decisions of surrounding agents, applying the same ego action would result in various future outcomes in different scenes. For instance, a turning action might lead to a collision in one scene but no collision in another. This characteristic further boosts the diversity of action-state pairs.
> > >
> > > We agree that introducing more patterns is a promising direction, and we will consider using LLMs for automatic pattern generation to cover more rare and critical ego actions.
> > >
> > > > ### Noise issue.
> > >
> > > Thanks for your clarification on the question. In most cases, our model does not exhibit CARLA artifacts in its visuals or dynamics. This is because the model is conditioned on real-world history frames and must generate a future that is visually consistent with that context.
> > > However, we have observed a limitation in extreme night scenes with low visibility, where predictions can occasionally degrade. This occurs because night scenarios are more frequent in our CARLA dataset than in our real-world data, and the lack of visual cues in the dark context can lead to hallucinations.
> > > There are two promising ways to address this issue:
> > >
> > >
> > > * Data Re-sampling: Increase the sampling rate of real-world night cases during training to better balance the data distribution.
> > >
> > > * Applying an upgraded simulator: Adopt the newest CARLA version (0.10.0 with Unreal Engine 5.5, compared to currently-used CARLA 0.9.15 with Unreal Engine 4.26). Its improved visual and physics engines will further close the simulation-to-reality gap.
> > >
> > > We will add more visualization and discussion to our failure mode analysis and future work sections.

---

> > > > ### Comment · Reviewer_SA1m · 2025-08-03
> > > > **Thanks**
> > > >
> > > > Thanks for these comments. These are very helpful to inform the method and the associated results. I have updated the rating to accepted.

---

> > > > > ### Author Response · Authors · 2025-08-03
> > > > >
> > > > > Thank you for the kind response and suggestions for improving the work! We will integrate our discussions into our revision.

---

### Official Review · Reviewer_J5UG · 2025-07-02

**Clarity:** 3
**Significance:** 3
**Originality:** 3
**Rating:** 5
**Confidence:** 3

**Summary:**

This paper proposes ReSim, a novel diffucion transformer world model method for self-driving that integrates expert real world driving data and simulation data for extreme edge cases not observed often enough in real data. The authors also proposed modifying noise sampling in diffusion, and incorporating an additional latent dynamics loss. The resulting model then can be more reliable in generating outcomes of suboptimal actions, making it excel at trajectory prediciton, planning, and visual simulation.

**Questions:**

- Q1: How do you pick the evaluation trajectories for human study?
- Q2: Could you provide quantitative results of ablations, as opposed to just image generations?
- Q3: Could you explain in more detail how you collect exploratory data in carla? You say that you have a pre-defined set of action patterns you can execute, what do they correspond to? Do you just let the PDM-Lite execute for a few steps, then interrupt and execute random actions? Or is there something else going on?

**Ethical Concerns:**

["NO or VERY MINOR ethics concerns only"]

**Final Justification:**

I believe this paper should be accepted as it's well-written, and provides an example of successful integration of simulation data to improve a self-driving system. This is critical for self-driving, and for other fields as well, making this paper valuable to the community.

**Limitations:**

yes

**Paper Formatting Concerns:**

no concerns

**Quality:**

3

**Strengths And Weaknesses:**

###### Strengths
- The paper is clearly written, and was overall a pleasant read;
- I like how this method is able to transfer the knowledge from Carla simulator and enable improved performance on real-world data such as Waymo and nuScenes.
- The experiments are thorough and showcase the strength of the method. I particularly appreciate the conducted human study.

###### Weaknesses
- I found the ablation study underwhelming, with noise sampling and dynamics consistency ablations only showing the qualitative results.

---

> ### Author Rebuttal · Authors · 2025-07-30
>
> Thanks for your valuable feedback. We respond to each of your concerns below and will incorporate the discussions into the revision.
>
>
> > ### Quantitative results on the ablation study of unbalanced noise sampling and dynamics consistency loss.
>
> We provide a quantitative ablation on the nuScenes dataset to validate the effectiveness of our proposed methods.
>
>
> |  Model             |    nuScenes FVD ( &darr; )    |
> |:-----------------------------|:------|
> | Baseline                     | 93.7  |
> | + unbalanced noise sampling  | 85.2  |
> | ++ dynamics consistency loss | 81.6  |
>
> The results show that both unbalanced noise sampling and dynamics consistency loss improve video prediction fidelity. Note: to make an efficient comparison, these ablation models were trained with less compute than our final model in the main paper.
>
> > ### How to pick the evaluation trajectory for the human study?
>
> For the human study, we created challenging test cases by pairing 20 random scenarios from the unseen Waymo validation set with mismatched, non-expert trajectories. For each sample, the associated trajectory is sampled from the same validation set, but with a different driving command. For instance, we might associate a "driving straight" scenario with a "turning left" trajectory as its non-expert action. We will clarify this process in the revised paper.
>
> > ### More details on CARLA data collection.
>
> Our CARLA data collection follows a structured "expert-takeover" process. First, the expert PDM-Lite policy drives for the initial period (1s). Then, control is switched to one of the following exploratory strategies to generate diverse, non-expert actions for 4s:
>
> * Slight Turns: The vehicle steers slightly left or right towards a randomly chosen angle between 10-20 degrees and then continues forward.
>
> * Hard Turns: The vehicle performs a sharp left or right turn towards a random angle between 40-60 degrees.
>
> * Forced Lane Changes: The vehicle executes a hard lane change to the left or right.
>
> * Tailgating: The vehicle disables its brakes and applies full throttle to induce a rear-end collision with a vehicle ahead.
>
> Each strategy also includes randomization of its internal parameters (e.g., turning angle, speed) to encourage action diversity. We will add these details to the revision. We will include the above details in the revision.

---

> > ### Comment · Reviewer_J5UG · 2025-08-03
> >
> > Thank you authors for authoring my questions. The details on Carla data collections clarify my doubts, please include these details in your manuscript.
> > I will keep my rating unchanged (5).

---

> > > ### Author Response · Authors · 2025-08-03
> > >
> > > Thank you for your thoughtful suggestions. We will include related discussions in the revision.

---

### Official Review · Reviewer_zQQ2 · 2025-07-04

**Clarity:** 3
**Significance:** 3
**Originality:** 3
**Rating:** 5
**Confidence:** 3

**Summary:**

The authors propose a method to make world model predictions more physically consistent with the given action input. The contributions appear to be training on CARLA sim data with randomized driving signals, as well as a motion consistency loss and a novel noise sampling technique for the underlying diffusion model. The method achieves strong results on benchmarks.

**Questions:**

Controllability: a) In the second-to-third image transition of Fig.1b, it seems like your model also went forward instead of turning more left? Was the action turn left during the whole sequence or just the first image? b) It is difficult to get an understanding of how controllable your WM is as the Trajectory Difference metric is I think an L2 loss over only a 2s prediction window. It would have been helpful to include some data on the drift in meters over longer predictions (the IDM itself has drift so it's imperfect but better than nothing).

Other agents: This uses randomized driving signals mainly for the ego vehicle, but what about the other agents? in the attached video nuscenes1 (nice that this was included!) it seems like driving forward makes it so the car in front never tried to stop when it otherwise actually did. Am I correct in assuming that the "distort reality to a nominal scenario" problem of WM prediction that you tried to address for the ego motion is still present when it comes to how other cars behave? I think you might be hinting at this problem in your limitations section but can you be a bit more clear?

CARLA artifacts?: Since the non-nominal situations are from CARLA, do you see the WM predictions looking more like CARLA / low-fidelity the longer you predict or the more you deviate?

**Ethical Concerns:**

["NO or VERY MINOR ethics concerns only"]

**Final Justification:**

The authors resolved my lingering concerns on artifacts and controllability. My question about other agents was a delimitation they might explore in future work.

**Limitations:**

Mostly there, see comments above.

**Paper Formatting Concerns:**

minor: Not sure if your terminology of "non-expert" is a good label in images for trajectories that I think are actually randomized (and in the examples consistently crash)?

**Quality:**

4

**Strengths And Weaknesses:**

Strengths:
- The paper addresses an important problem with physical consistency of world models
- The paper is well written, I greatly enjoyed reading it.
- Seems to have strong results on benchmarks (e.g. NAVSIM)

Weaknesses:
- Training on sim data with randomization seems rather obvious and also with possible obvious downsides that could have been explored better. If you deviate from the nominal state of real/expert driving data and benefit form this sim training, do you also visibly see the scene turning more into something that looks like CARLA?
- Some slight overclaims in the writing: E.g. "we are the first to demonstrate that integrating both sources overcomes the shortage of unsafe driving behaviors in real-world data". "Overcomes" is a strong word considering that it still does not seem fully reliable, especially in the behavior of the other traffic agents. This does not seem like a solved problem. Consider softening this.

---

> ### Author Rebuttal · Authors · 2025-07-30
>
> We appreciate your insightful review and have carefully considered your comments. We will revise the manuscript based on the discussions below.
>
> > ### Visual artifacts brought by CARLA data.
>
>
> Thanks for the question. We typically do not see CARLA artifacts because our model is grounded in real-world visuals. The model is first pre-trained on the massive, real-world OpenDV dataset, establishing a strong visual generalization. When simulation data is introduced in later stages, the model learns to predict dynamics while preserving the realistic visual style learned from the input context.
>
> However, in extreme night scenes with low visibility, which are more frequently shown in our CARLA data than in our real data, the model can occasionally struggle. One possible way to mitigate this issue is to increase the sampling rate of real-world night cases during training, e.g., repeating this data multiple times.  We will add the discussion and more visualization to our failure mode analysis.
>
> > ### Slight overclaim in "overcome".
>
> Thanks for this suggestion. We will revise "overcome" to a softer claim such as "alleviate".
>
>
> > ### Clarification on controllability in Figure 1b.
>
>
> In Figure 1b, the control trajectory, which is applied to the following future sequence, is a slight left curve to the sidewalk. ReSim correctly follows this non-expert action, while the baseline continues driving straight. We amplified the curve in the diagram for illustration only.
>
> For your reference, the actual control trajectory is "[(7.52, 0.09), (15.20, 0.48), (22.97, 1.17), (30.58, 2.13), (38.25, 3.39), (45.80, 4.93), (53.25, 6.76), (60.62, 8.86)]", representing the relative displacements of 8 positions over the next 4 seconds. The first element of each coordinate indicates the forward displacement, and the second represents the lateral displacement.
>
> We will update Figure 1b in the revision to more precisely depict the control trajectory.
>
> > ### Trajectory difference with challenging actions in a longer horizon.
>
> To validate the controllability of ReSim on non-expert actions with large drift from recorded action, we construct a subset by associating each scenario with a challenging, non-expert action sampled from other scenes. The average end point distance of this subset is 24 meters, and the evaluation horizon is 4s. The quantitative results below show that including simulation data improves the controllability of non-expert actions with large drift.
>
> |  Model             |   Trajectory Difference ( &darr; )  |
> |:-----------------------------|:------|
> | ReSim (w/o sim.)  |  5.32  |
> | ReSim                   | 3.92  |
>
> Note: The Inverse Dynamics Model (IDM) used for this metric is trained only on expert data, so its accuracy on these hazardous videos is limited, as you mentioned. Vista is not included as it cannot be controlled on a 4-second horizon, which exceeds its training window.
>
> > ### How does the model handle the behavior of other agents?
>
> Since our work focuses on controlling ego vehicle only, the behavior of surrounding agents is learned implicitly from all training corpus, including OpenDV, NAVSIM, and CARLA, and cannot be explicitly controlled during inference time. Indeed, our CARLA data also introduce "non-expert" behaviors for surrounding agents since they are governed by various behavioral strategies such as being aggressive. Scaling up this simulated data to further improve stochastic modeling of all agents is a promising direction, and we will add this point to our future work section.
>
> > ### Terminology of "non-expert".
>
> You are correct that our "non-expert" actions are generated using randomized strategies to ensure action diversity . We use this term to align with robotics literature [1, 2], where "non-expert" commonly contrasts with "expert demonstrations" and refers to a broad set of suboptimal or exploratory behaviors. We will clarify this in the revision.
>
> ---
>
> [1] Pre-training with Non-expert Human Demonstration for Deep Reinforcement Learning
>
> [2] Reward-Conditioned Policies

---

> > ### Comment · Reviewer_zQQ2 · 2025-08-02
> > **Reply to authors**
> >
> > I thank the reviewers for their explanations. I am satisfied with my Accept rating.
> >
> > One minor comment on the "non-expert" terminology when you mean automatically generated (mostly random) driving trajectories, unless this has been used by previous AD papers, consider clarifying further in the paper what you mean by this. I was initially not sure if this referred to some kind of low-quality human data collected in sim (not every human is an expert at driving, some lack driver's license, while some are professional drivers or compete - real or video game driving).

---

> > > ### Author Response · Authors · 2025-08-03
> > >
> > > Thanks for the constructive suggestion. We will clarify the term "non-expert" in the revision to make it easy to understand and avoid misunderstanding.

---

### Decision · Program_Chairs · 2025-09-17

**Decision:**

Accept (spotlight)

**Comment:**

The paper presents ReSim, a controllable diffusion-transformer world model for autonomous driving trained on heterogeneous real expert, simulated expert, and rule-based non‑expert CARLA data, plus a Video2Reward module. After rebuttal, all reviewers agree that the paper addressed the concerns. The final suggestions is Accept.